# Assessing Systematic Blade Production in the Indian Subcontinent with Special Reference to Eastern Gujarat

**Gopesh Jha** [1,2,*][ID]**, Vidhi Kothari** [2]**, Varun Vyas** [3] **and P. Ajithprasad** [2]

[1]  Department of Archaeology, Max Planck Institute of Geoanthropology, 07745 Jena, Germany
[2]  Department of Archaeology and Ancient History, The Maharaja Sayajirao University of Baroda, Vadodara 390002, India
[3]  Archaeological Material Sciences, Universidade de Évora, 7005-869 Evora, Portugal
*   Correspondence: gjha@shh.mpg.de

**Abstract:** Blades as a component of lithic assemblages hold significant importance to understanding the more recent part of human evolution, particularly with regard to the emergence and adaptations of *Homo sapiens*. The systematic production of elongated stone blanks provides several advantages, including a longer cutting edge and high efficiency in raw material utility. However, the reasons behind the development of these technological forms and the chronological patterns of systematic blade production remain poorly understood in many regions, despite a clear overall intensification in the Late Pleistocene. The South Asian Paleolithic archive is full of blade-bearing assemblages, most of which are defined as Upper Paleolithic or Late Paleolithic. However, many of these previously assumed 'Upper Paleolithic' tool components prominently appear in Middle Paleolithic contexts. Here, we discuss some of the most recent case studies of blade-bearing assemblages from Eastern Gujarat that show an in situ emergence of blade technology from advanced Middle Paleolithic technology, suggesting localized origins of blade technology.

**Keywords:** blade technology; Middle Paleolithic; Upper Paleolithic; convergence; transition





## 1. Introduction

Stone tools are the most abundant form of evidence used to understand past human behavior [1–4]. Changes in tool size, design, and assemblage composition are often used to suggest the occurrence of a behavioral shift, including between different hominin species [5,6]. The emergence of blade technology is one such behavioral change that has been focused on as a potential indicator of changes in hominin populations, adaptive response to climate change, or socioeconomic strategies [4,7–11]. In particular, in many regions, the appearance of blades has been associated with the arrival of *Homo sapiens*, with contrasts being drawn with previous hominin lithic assemblages [2,4,8,12,13]. The same is true of South Asia, where the appearance of blade technologies (~MIS 3) has been closely associated with dating the appearance of our species in the region, being used to support or refute dispersal hypotheses [14–18]. On the other hand, research in South Asia has highlighted the way in which blade technologies may have emerged from in situ lithic technologies, indicating greater longevity of our species and potentially its interaction with other hominin populations [17–22].

## 2. Indian Upper Paleolithic: Result of Colonial Systematics

From the time of Bruce Foote, Cammiade, and Burkitt to the times of M. L. K. Murty and V. N. Misra, many scholars have reported and worked on several blade-based sites in South Asia (Figure 1), traditionally defined as Upper Paleolithic [23,24]. However, evidence of these assemblages is sporadic in comparison to the succeeding and preceding industries. The Indian Upper Paleolithic (UP) is conventionally represented by flake-blade, blade, and burin-based assemblages [5]. Sali was the first to establish the stratigraphical context of

Indian UP assemblages at the site of Patne [6]. The UP horizon at Patne is sandwiched between Middle Paleolithic (MP) and Mesolithic horizons. The Patne UP shows two-fold occupation wherein the later of the two incorporated the evidence of behavioral modernity (ornamental objects), which is absent in most of the South Asian UP assemblages [16]. The understanding of South Asian UP industries is therefore often disputed, on both local and regional scales, and there appears to be significant variation in terms of composition and technological affinity. This ambiguity is mainly due to the lack of systematic technological assessment and chronometric dating.

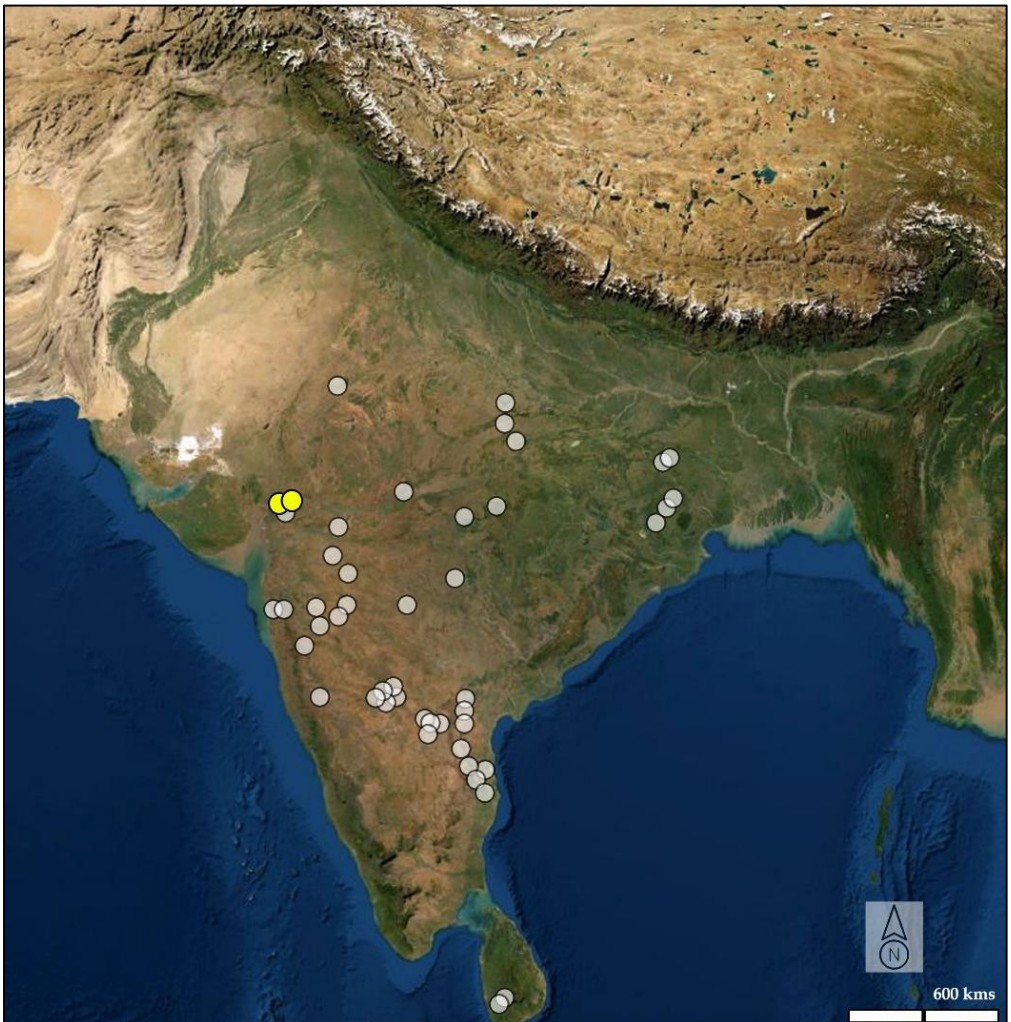

**Figure 1.** Major Upper Paleolithic sites in India (adapted from Murty, 1979 [24]). White dots represent UP sites of Indian Subcontinent. Yellow dots represent the Jogpura and Bhuvera. (Base map source: Nat Geo).

Previous assumptions of the rapid emergence and expansion of these flake-blade and blade artefacts and the technological shift have also now been called into question by studies which suggest a more gradual appearance [17,18,21]. Many 'Upper Paleolithic' tool components have also been found within MP contexts—for example, burins, end-scrapers, and microblades are found within MP layers at Bhimbetka [16]. Rather than considering them as a distinct techno-complex, blade technologies may therefore represent a continuation of MP technological practices in different parts of South Asia [20]. Similar trends were noticed at the site of Jwalapuram [18]. Here, Clarkson et al. demonstrated the in situ occurrence of the blade and microblade technologies in the latest MP sites and Levallois/discoidal reduction methods in the Late Paleolithic assemblage (JWP 9D), showing

the localized nature of technological transition. Clarkson argues that the backing of blade blanks and their miniaturization appear gradually, with the blade size (length) decreasing moderately, demonstrating the gradual, local emergence of the blade-based industries (referring to the Late Paleolithic techno-complex) rather than rapid replacement [18,20].

Similar trends in blank production have been observed at different blade-bearing assemblages in South Asia. As Clarkson highlighted, most of the MP-associated blades are longer than 4 cm [18,20]. By contrast, the length of the blades (i.e., microblade and microliths) of LP assemblages is smaller than 4 cm. Figure 2 represents a comparison of the length of blades from 10 UP/MP and 5 LP assemblages. Most of the blade-bearing UP/MP sites in Figure 2 are undated, however. Figure 2 shows the distinct grouping between UP/MP blade assemblages and Late Paleolithic (LP) blade assemblages, which points towards the distinct nature of UP/MP blades and highlights the shifting preference in blank size. The length of blades from some of the previously defined UP assemblages, such as Budha Pushkar (all layers), Baghor, and Renigunta upper layer, overlap with the length of Late Paleolithic/microlithic artefacts. Blinkhorn [25] argues that sites such as Budha Pushkar are misidentified as UP, as they yield strong characteristics of the microlithic techno-complex. Indeed, many Paleolithic sites are described as UP assemblages based on typological assumptions.

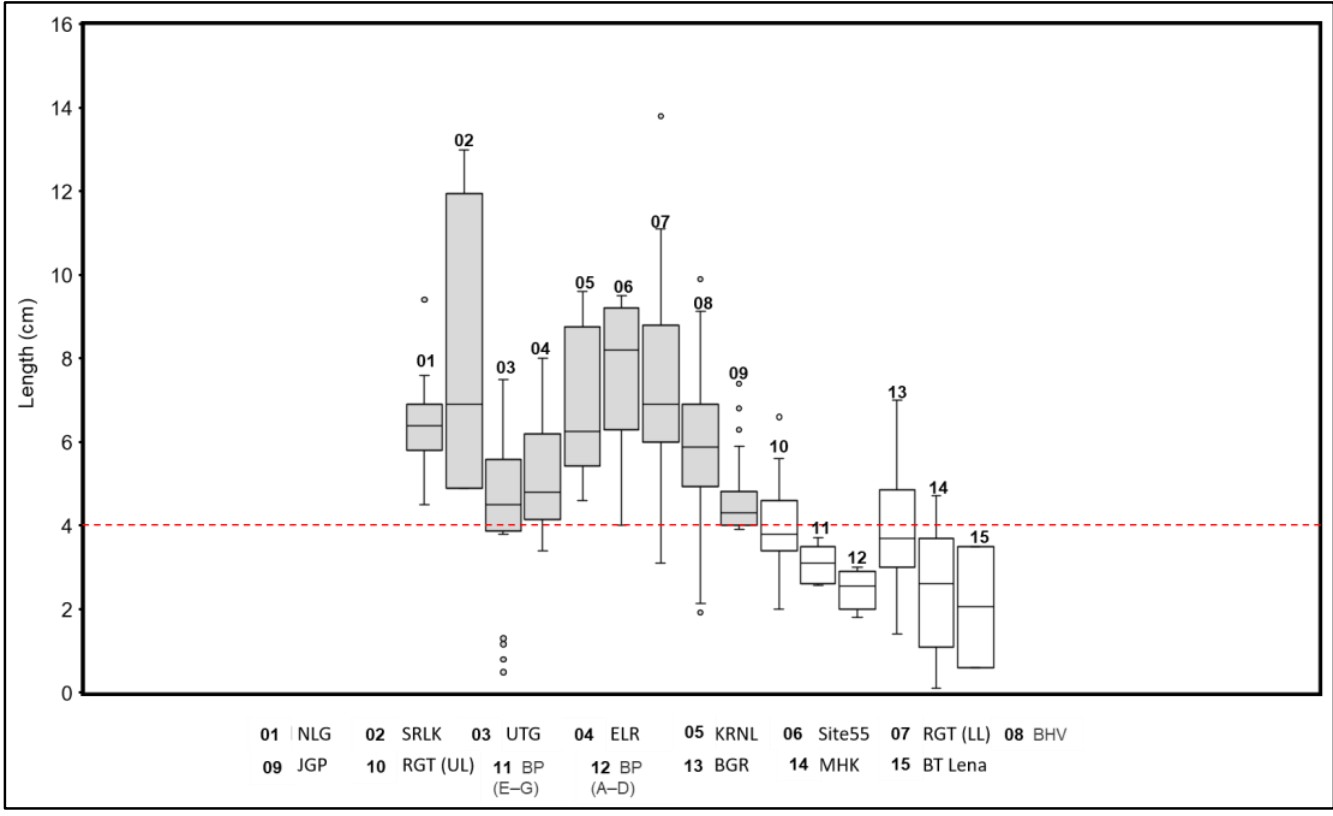

**Figure 2.** Comparison between the length of blades from 15 sites. UP/MP sites (Nalgonda (NLG), Srikalahasti (SRLK), Upper Tungabhadra (UTG), Eluru complex (ELR), Kurnool complex (KRNL), Riwat Site 55, Renigunta lower level (RGT LL), Bhuvera 3 (BHV), Jogpura (JGP), Renigunta upper level (RGT UL), Budha Pushkar (BP E–G), Budha Pushkar (BP A–D), and Baghor (BGR)) and LP sites (Mehtakheri (MHK) and Batadombalena (BT Lena)). Dotted line marks the length of 4 cm. All blade assemblages from MP or UP context are larger than 4 cm, and all LP blades are smaller than 4 cm. Two distinct size groups within blade assemblages reflecting their distinct technological affinity (i.e., MP and LP).

Overall, it must be taken into account that Paleolithic archaeology (Figure 2) in the Indian subcontinent has been heavily influenced by European technological framework and nomenclature. For instance, Foote used the term 'Magdalenian' to define the bone-tools of Kurnool [24]. Often, conventional European terms are misleading and fail to adequately define the various aspects of Indian prehistoric records. Of late, scholars have used the term "Late Paleolithic (LP)" to describe these assemblages [16]. Although the Late Paleolithic put more emphasis on the microblade/microlith industry of the Late Pleistocene (as different from later microlithic assemblages), it often neglects the nature of macro-blade (larger than 4 cm) components, which flourished parallel to the late MP techno complex. Thus, it becomes difficult to define the technological affinity of these large blades, i.e., whether one should see them as a separate technological entity or treat them as a variant within the broader umbrella of MP technocomplex. The current paper discusses evidence of the parallel occurrence of systematic blade production within MP contexts from two sites in Eastern Gujarat to understand the technological features of early blade-dominated assemblages.

### 3. Materials and Methods

#### 3.1. Research Area

Geographically, Eastern Gujarat is marked by the geographical division where the southern fringes of Aravalli intersect with the western spurs of Vindhya and form a rugged topography. The region is drained by two major river systems (i.e., Mahi and Narmada). Climatically, it spans sub-humid to semi-arid categories and is known for its thick, dry, deciduous forest, which acts as a habitat for a large variety of wild fauna. The rugged topography provides a key watershed, providing long-lasting resources of water (the Orsang and Sukhi valleys). One can also see this region as an eco-hotspot or refuge due to its rich ecological resources, something which may have applied to the past given its Paleolithic record of hominin occupation [26–28]. The geographical corridor of Eastern Gujarat likely played a significant role in early human dispersal [17]. As far as UP sites in Gujarat are concerned, there is a single UP site in this vast geographical zone. The Cambridge-Baroda University expedition of Allchin and Goudie (1971–1974) led to the discovery of the UP site at Visadi in the Orsang valley [29]. Nevertheless, Visadi's status as a UP site is highly debated by several scholars for a number of reasons [27]. The site of Jogpura (JGP) in the Sukhi valley and Bhuvera in the Panam valley also yield a Visadian kind of assemblage showing evidence of systematic blade production (current work). Here, we describe the technological characteristics of blade production at the formerly mentioned sites to highlight their similarities and differences.

#### 3.2. Lithic Analysis

An elaborate morphometric analysis was carried out to test different aspects of JGP blade assemblages, such as knapping control, morphological variability, degree of reduction, and typology, to see if they coincide with the associated features of systematic blade production. The blades in the current study are classified based on the major attributes described by Bordes [15]. Different morphological attributes (i.e., length, width, thickness, angle of lateral margin, no. of parallel-subparallel scars on the dorsal face of the blanks, etc.) of blades were manually recorded using a caliper. The current analysis intends to understand the factor of control in terms of knapping, which must have led to morphological variability or standardization. We seek to determine if stone tool knappers made these tools highly uniformly, suggesting very specific needs in terms of size and shape, or whether they produced them more casually, resulting in high variability. A broader goal is to understand the technological affinity of assemblages such as JGP and BHV that are highly hybrid in nature.

## 4. Results

*4.1. Jogpura*

4.1.1. Geographical Settings

Jogpura (Lat. 22°28′0.51″ N; Log. 73°47′29.82″ E) is situated on the relatively flat land formed at the top of an extensive rocky ridge. The current region (i.e., north-east Gujarat) of the site is characterized by its rugged topography. Several Acheulian and Early and Late Middle Paleolithic localities were located on the top of high hills at the Kevada, Pani, and Jogpura (Figures 3 and 4). The constant weathering and erosion have turned this table-top land into an undulating landscape, with deep gullies, prominent rock promontories, and a relatively flat basin-fill with thin soil cover. The undulating top is the erosional remnant of the table-top land of Pleistocene origin. From the Paleolithic site of Bar, undulated hill-top stretches about 10 km towards the west with a maximum height of 350 m from the ground. Gradual erosion has also exposed multiple horizons of Paleolithic archives across the area. The continuous spread of the Acheulian and the Middle Paleolithic artefacts shows the richness of the site. The artefacts occur in discrete clusters, indicating their complex depositional nature.

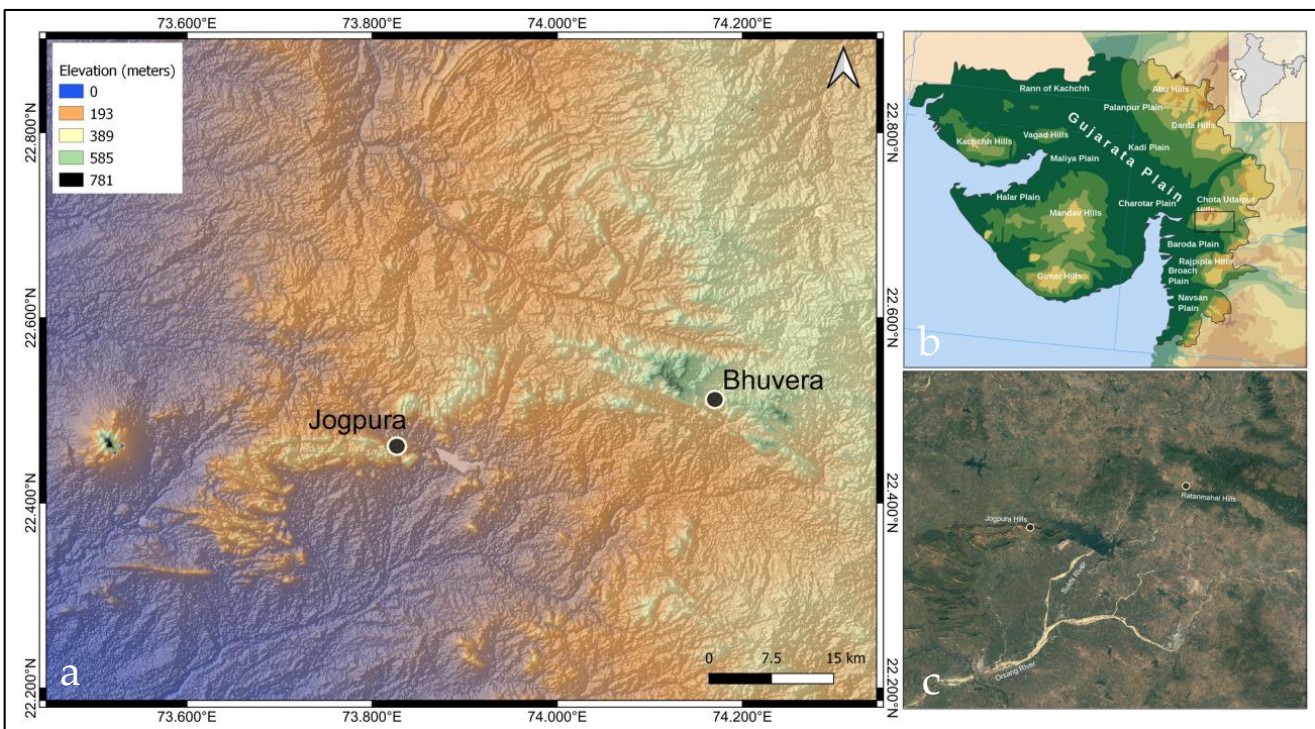

**Figure 3.** (**a**) Digital elevation map of research area: note the location of Jogpura and Bhuvera; (**b**) geographic map of Gujarat highlighting Chhota Udaipur hills where sites are situated (courtesy—Goran Tek); (**c**) LANDSAT of research area (source—Google Earth).

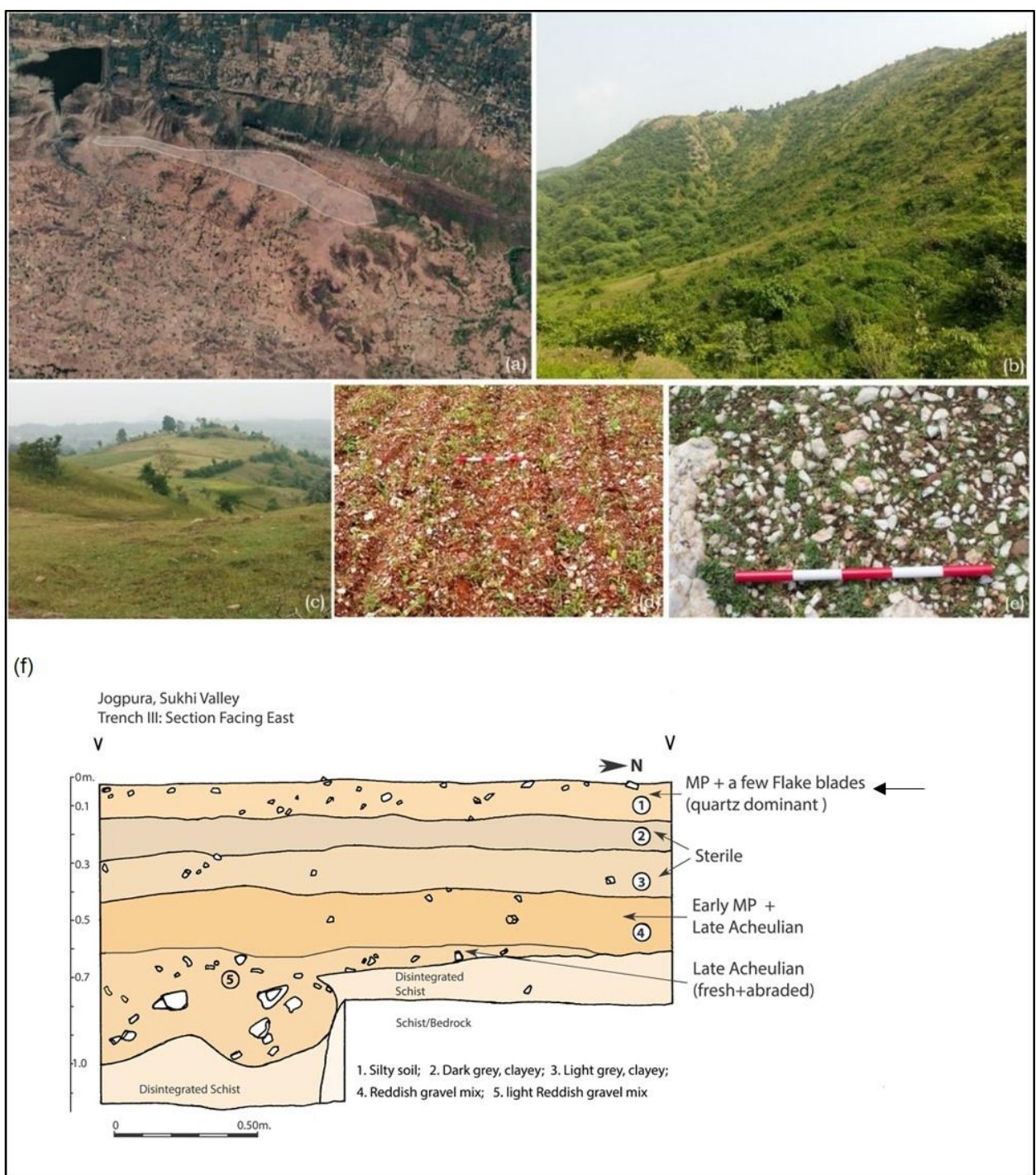

**Figure 4.** (**a**) Satellite image of table-top land of Jogpura; (**b**) distant view of Jogpura hill; (**c**) macro view of undulated table-top land; (**d**) vein quartz artefact cluster exposed by farming; (**e**) Intvada cluster of vein quartz artefacts.; (**f**) stratigraphy of Jogpura Site—black arrows point towards the stratigraphic location of quartz-dominated MP assemblage.

### 4.1.2. Stratigraphy

The stratigraphic contexts of the Paleolithic materials at Jogpura were revealed from the three trenches excavated at the site in 1994 [27] by exposing multiple horizons of Paleolithic occupation (Figure 4), ranging from the late Acheulean to the MP techno-complex [26,27]. The presence of early MP and the preceding late Acheulean assemblage at the site provides a unique opportunity to understand the emergence of prepared core

technology there. Here, we focus on the upper-most Paleolithic occupation at the site, which also shows the first evidence of systematic blade production (Figure 4). These are found as discrete surface clusters of quartz-based, flake-blade-dominant assemblages eroding out of light-brown silty-sandy sediment context. Interestingly, all the above lithic clusters are situated next to the exposed quartz vein outcrops. The 1994 JGP excavation also confirms the surficial context of these quartz-based assemblages (Figure 4). Despite being a surface assemblage, we did not observe any sign of inter-assemblage mixing because the assemblage was entirely composed of vein quartz, whereas the preceding assemblage was predominantly composed of quartzite. The complete shift in raw material from the preceding occupational layer is a remarkable feature.

One of these quartz artefact clusters was targeted, and artefacts were collected by using a radial collection strategy in order to record the spatial distribution of the clusters. Artefacts are plotted on the graph to obtain a better picture of the artefact distribution in the clusters and determine the rate of disturbance that must have been caused by anthropogenic alteration. Overall, the artefact cluster shows limited spatial movement, although some of the artefacts in the peripheral region may have suffered displacement due to the steep slope on the northern edge of the cluster. However, it is inappropriate to read any significant behavioral signature in the documented spatial distribution due to severe subsequent agricultural (land tilling)-induced disturbance at the site.

### 4.1.3. Composition of the Cluster

The JGP cluster is one of the richest lithic clusters on the eastern slopes of Jogpura hill. It contains a huge amount of shattered quartz clast—most of them have flaking marks. The opportunistic behavior of knappers resulted in the heavy production of waste (debitage). The JGP cluster is predominated by blade blanks. However, there are a remarkable number of diagnostic PCT (prepared core tech.) products (Figure 5) that show diverse technological manifestations. A total of 385 vein quartz artefacts were collected from the JGP cluster, which are broadly classified into two categories: core and flake. The cluster mostly comprises blade, burins, denticulates, notches, scrapers, diminutive bifaces, and cores (unidirectional single platform cores as well as PCT cores). Many blades, as well as other lithic components, are broken, mostly due to agricultural tilling. Here, we focus on a small dataset, which only discusses the details of blade artefacts. A detailed classification of the JGP cluster is explained in Table 1.

**Table 1.** Summary of artefacts collected from JGP cluster.

| Artefact Type | | Quantity |
|---|---|---|
| Core | Blade core | 38 |
| | Levallois core | 2 |
| | Radial/discoidal core | 27 |
| Splits | Blade | 252 |
| | Flake | 58 |
| **Details about blade** | | |
| | Complete blade | 172 |
| | Broken blade | 80 |
| | Retouched blade | 48 |
| | Blade with faceted platform | 14 |
| | Blade cross-section (triangular-trapezoidal) | 152:99 |
| | No. of dorsal arises (00-01-02-03) | 19-152-78-02 |

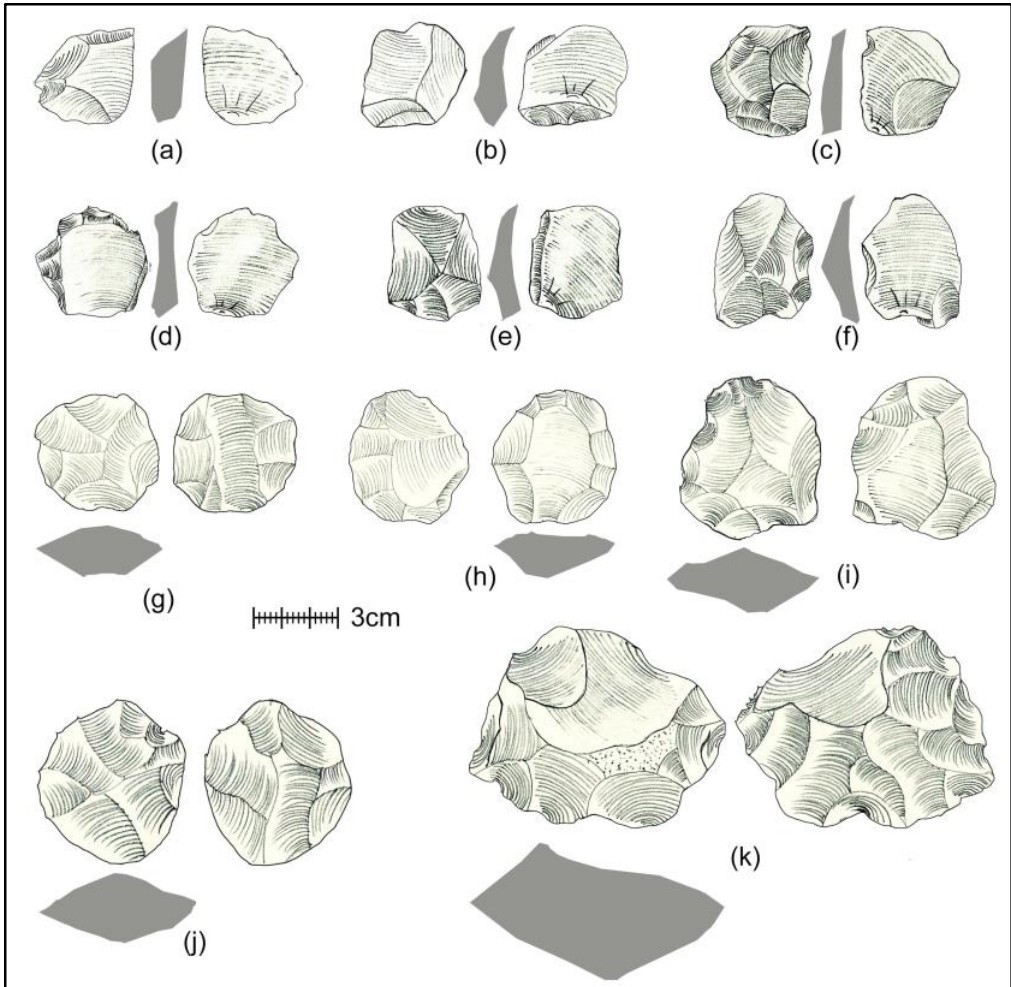

**Figure 5.** Vein-quartz-based prepared core tech. components from JGP cluster. (**a–e**) PCT flakes, (**f**) reworked flakes (bifacial working on lateral margins), (**g–k**) radial cores. All of these cores are composed of vein quartz.

### 4.1.4. Raw Material

The site has a rich source of quartzite, which was apparently heavily exploited by the Acheulean and MP tool makers. Interestingly, the succeeding blade-dominated MP industry is completely based on vein quartz, which is found within the gneiss formation (outcrops are seen all around the hill-top). Quartz assemblages generally appear to be comprised of amorphous and shattered clast, which are not easily recognizable as humanly modified. Due to the unpredictable conchoidal fracture mechanism of quartz, it is hard to understand the reduction patterns. The typological classification of quartz-based lithic industries is therefore extremely difficult [30]. The same factors also make quartz a potentially inferior raw material. There are multiple clusters of quartz artefacts spatially spread all along the Jogpura hill. Most of the clusters are situated close to exposed quartz veins. The selection of vein quartz over fine-grained quartzite at Jogpura is an issue that defies immediate answer.

### 4.1.5. Reduction Pattern

It is clear that the knappers of JGP were well-versed in prepared core technology. Higher abundances of residual components (core and debitage) in the JGP assemblage demonstrates the heterogenous nature of MP technology. The assemblage has a wide va-riety of cores, which include 27 discoidal, 2 Levallois, and 38 unidirectional blade cores. Unidirectional flaking is predominant in the JGP assemblage, which is often considered a major characteristic of the blade industry (Figure 6). Tabular or sub-tabular quartz clasts

with nat-ural straight ridges were preferred for knapping. The natural straight ridges were primarily exploited to produce initial elongated blanks. This is followed by the recurrent removal of blades using arises of primary removal. Most of these blade cores have two main components, i.e., broad and flat sub-rounded platform and reduction face (showing recurrent removal of elongated blanks). The flake removal count for most of the single-platform unidirectional cores is highly inconsistent, reflecting minimal removal (less than 3) in most cases. There is a variety of tabular and sub-tabular vein quartz waste all around the site. This also points towards the maximum exploitation of raw materials or the opportunistic behavior of the knappers.

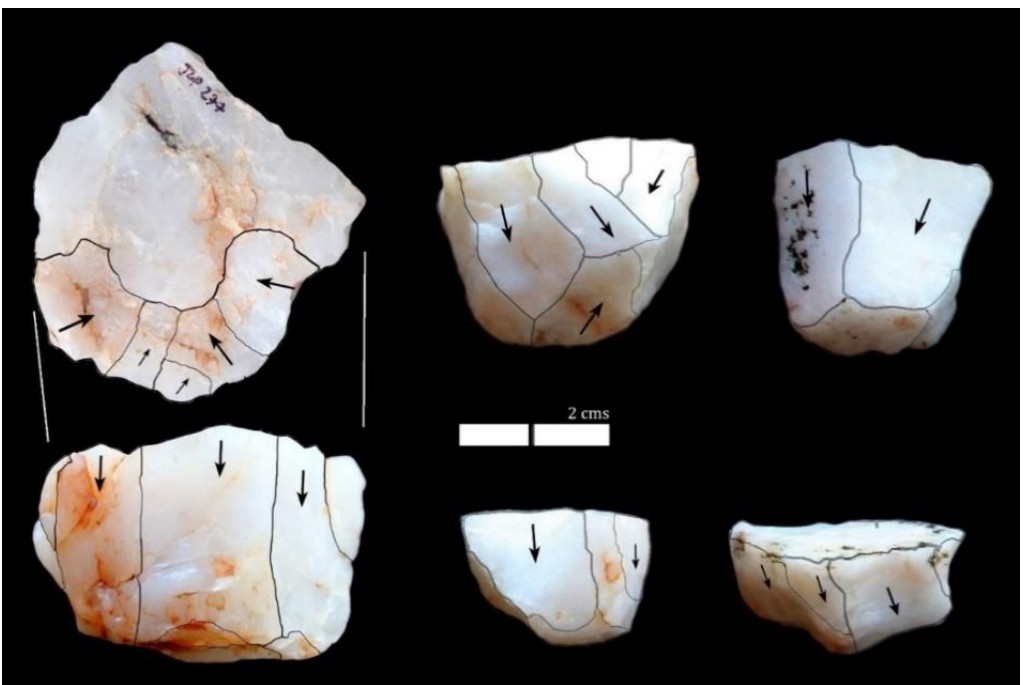

**Figure 6.** Single platform unidirectional blade cores from Jogpura.

### 4.1.6. Morphological Variability

Blade technology is known for its highly standardized blank production and raw material efficiency by the systematic preparation and maintenance of core volume [31]. Hence, it is important to assess the morphological variability in the current assemblage. Morphological attributes such as length, width, thickness, elongation (length:width), and flatness (width:thickness) were recorded to understand the variability in size which may reflect the varying degree of control of the flaking technique, or either functional or cultural preferences [32]. Blades are defined by their length, which separates them from other blank types. The length of 252 blades ranges from 20 mm to 74 mm with a mean length of 37.23 mm (Figures 7–9). The most preferred length among the JGP blade assemblage is around 4 cm, which is comparatively bigger than known South Asian microblades. The mean width of JGP blades is 20.66 mm, where the width of 93.6% of the blades falls in the range of 15 mm to 30 mm. There are certain anomalies in length as well as width because of the presence of a few unduly large, elongated flake-blades. As far as elongation is concerned, the average elongation of the JGP blade is 1.92 mm, suggesting a fairly elongated assemblage.

Another component that defines a blade is the uniform thickness. Thickness is recorded in three parts (proximal, medial, and distal). The average thickness of the JGP blade assemblage is around 9.37 mm with a standard deviation of 3.92 mm. Comparison within proximal, medial, and distal thickness shows a higher degree of uniformity. This uniformity is also reflected in flatness measurement. The mean flatness of the JGP blades is around 2.17 mm with a standard deviation of 0.69 mm. All aforementioned measurements

show a higher degree of morphological standardization, which also reflects a higher degree of knapping control.

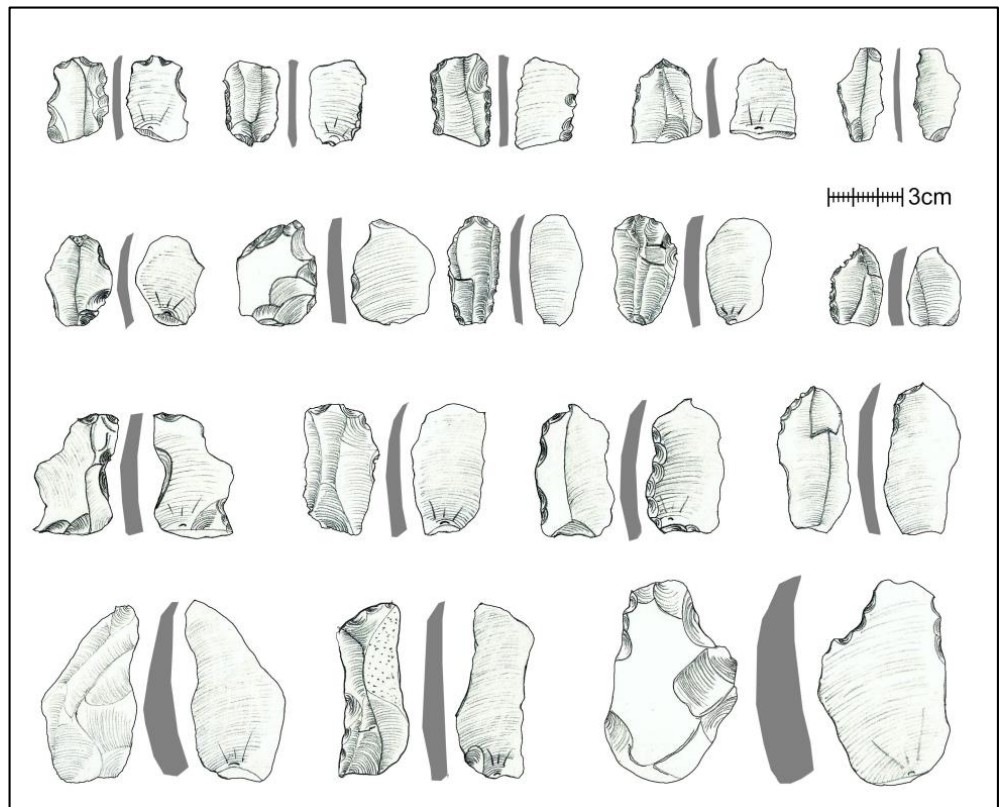

**Figure 7.** Blade artefacts from Jogpura.

4.1.7. Knapping Control

Blade techno-complexes are known for their higher degree of knapping control. The maintenance of parallel lateral margins is one of the most important aspects of blade technology, which reflects the knapping control. Standardization or variation in lateral margins can highlight the degree of control and precision in terms of blade production by recording the angle of lateral margins [32–34]. This attribute provides a measure of the degree to which flakes diverge or converge along their percussion axis. Negative angles indicate expansion of the lateral margins along the percussion axis away from the platform (divergence and convergence). A total of 189 out of 252 blades fall in the category of 85° to 95°, which certainly suggests a higher number of blades with a parallel–subparallel lateral margin (Figure 9). Hence, one can observe a higher degree of knapping control and precision in terms of blank production.

Apart from the angle of the lateral margin, it is extremely important to record guiding ridges (or arises), which are an essential marker for blade production, as they show the definite intention of a knapper who uses these straight or converging fracture path (plane) to detach longer and more parallel blanks [32,33,35]. The JGP assemblage has 152 blades with a single ridge, 78 blades with double ridges, 2 blades with triple ridges, and 19 blades with no ridge (most of them are primary removals). A pronounced increase in the number of straight parallel dorsal ridges on core and flakes suggests the recurrent production of these elongated blanks and points towards systematic blade production.

4.1.8. Other Technological Features

There are a few other prominent features of the JGP blade assemblage that are technologically important. There is minimal or no evidence of platform preparation in blade cores. Most of the quartz clasts have natural flat and perpendicular faces, which can be used for

blade production without any modification. As far as other modifications are concerned, most of the blades in the JGP assemblage do not show any major retouching or backing. However, most of them have use-marks/damage at the lateral margins. Some of these blanks are so exhaustively used that they are further converted into notches.

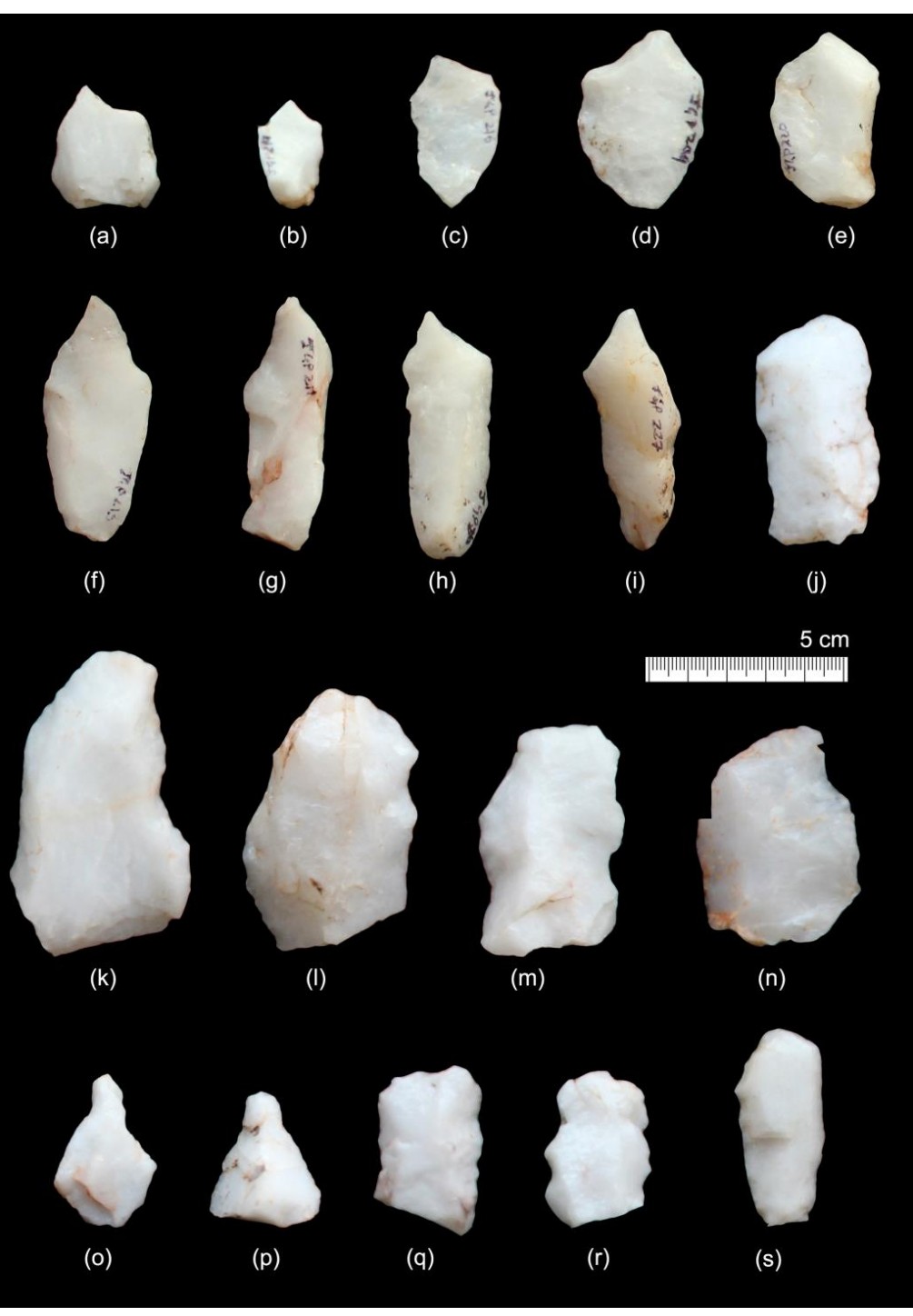

**Figure 8.** Diagnostic tool types from JGP. (**a–i**) burin, (**j,q,s**) side scraper, (**k–n**) notch, (**o,p**) awl, and (**r**) denticulate.

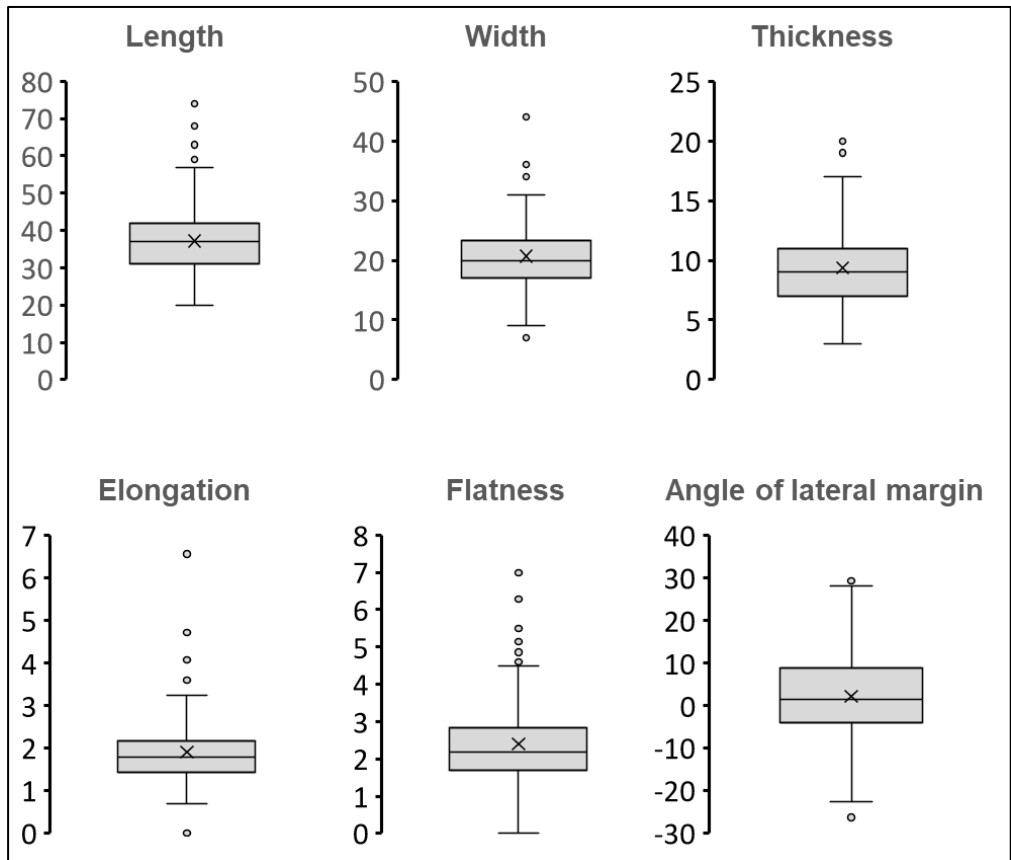

**Figure 9.** Boxplot showing range of different morphological indices of JGP blades.

*4.2. Bhuvera, Ratanmahal*

4.2.1. Geographical Setting and Nature of Artifact Occurrence

Ratanmahal is situated in Limkheda Taluka in the south-eastern part of the Panchmahal district of Gujarat in Western India. The area lies between 22°30′ and 22°40′ N and between 74°0′ and 74°12′ E. The hilly portion consists of a network of ridges that forms the rugged spurs of the Vindhyan range [36]. The discontinuous chain of hills is covered by a vast expanse of deciduous forest, which also marks the eastern border of Gujarat state. The undulating tabletop land in the high erosional zone of Ratanmahal is characterized by its deep gullies, narrow ravines, prominent rock outcrops, and relatively flat plains. The current area of research is part of the Sloth Bear sanctuary and comes under the jurisdiction of the State Forest Department. There are multiple hamlets/villages of tribal communities situated in different niches of the plateau (i.e., Pipargota, Alindra, Bhuvera, Kanjeta, etc.). The topography of Ratanmahal is similar to Jogpura. However, the plateau is much larger than Jogpura.

Bhuvera is one of the villages located on the southern periphery of the plateau. The site was discovered and documented by a team of postgraduate students from M.S. University, led by Ms. Vidhi Kothari. Multiple localities of Middle Paleolithic and Late Paleolithic/Microlithic have been found exposed in the site and modified by fluvial erosion. Three major localities (i.e., BHV 1: 22°31′39.65″ N 74°08′22.92″ E; BHV 2: 22°31′24.08″ N 74°08′38.25″ E; and BHV3: 22°31′21.52″ N 74°08′40.52″ E) were selected for further investigation (Figure 3). All these localities are predominated by Middle and Late Paleolithic/Microlithic components. Bhuvera 3 (BHV3) has a dense concentration of artefacts spread around a large field, which is exposed by farming as well as fluvial sheetwash (Figure 10). The lithic assemblage at BHV3 is predominated by a flake-blade component.

The assemblage also has a high number of PCT-based components (i.e., Levallois cores, radial/discoidal cores, PCT/Levallois flakes) (Figure 10).

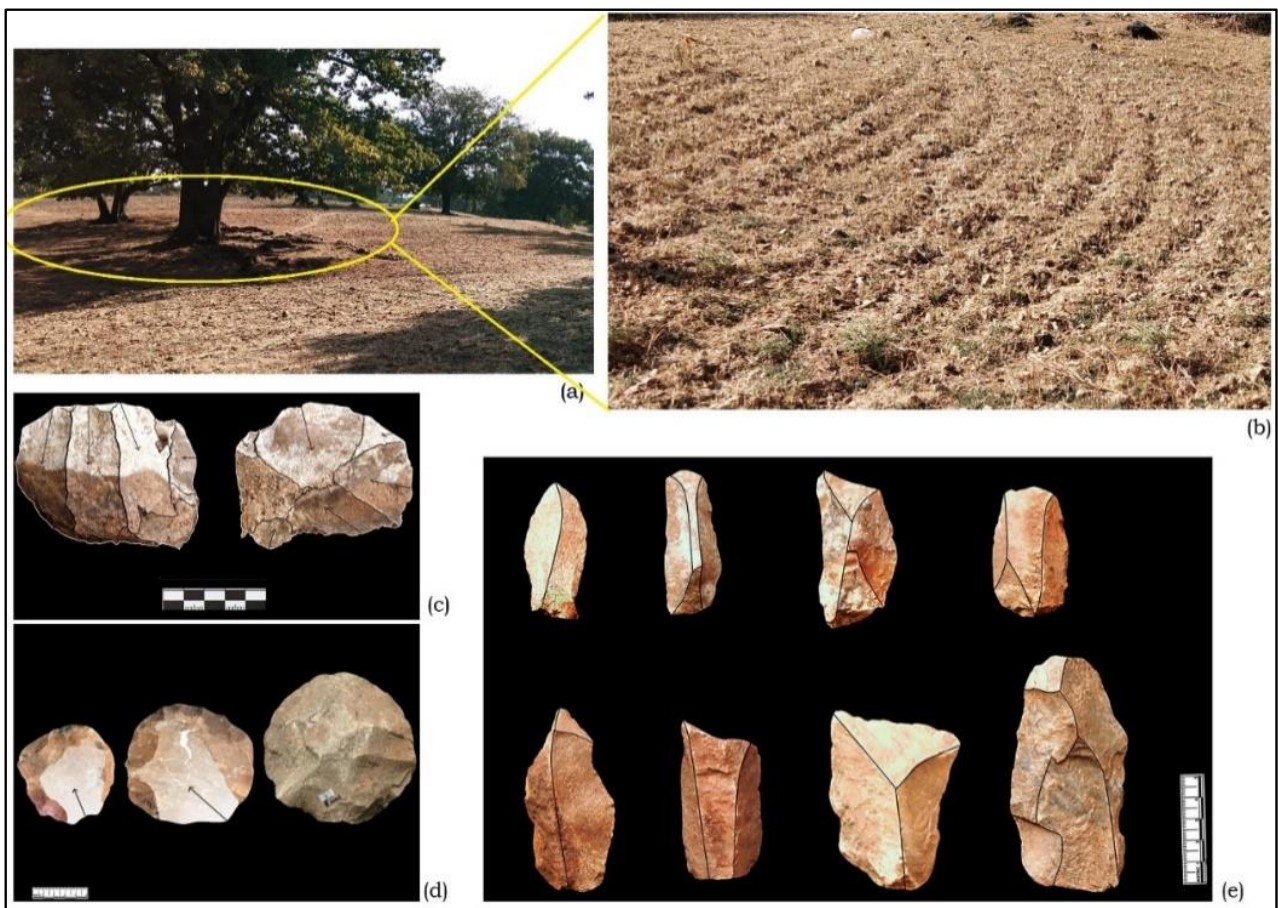

**Figure 10.** (**a**) Distant view of artefact-bearing field; (**b**) exposed artefact cluster due to farming; (**c**) blade core from BHV3; (**d**) Levallois cores from BHV3; and (**e**) blade-like blanks from BHV3.

The occurrence of these elongated blade-like blanks along with MP components makes BHV3 a suitable site for exploring the technological changes occurring in the later phase of Middle Paleolithic culture. The chronological status of the site is yet to be established, but it is quite certain that the lithic components of BHV3 have an affiliation with the Middle Paleolithic techno-complex. A total of 218 artefacts were collected during the transect survey (Table 2 and Figure 11). BHV3 has a scarcity of finished tool types and is heavily dominated by residual components (core and debitage), which denotes that the tools were primarily produced at this site. Most of the blanks have a high cortical area and seem to be primary removals (probably detached during the early stages of core preparation). However, the locality does not have any raw material sources in its close vicinity. Hence, it is inappropriate to make a conclusion, as the current study is based on a very small part of the area. Most of the adjoining areas are inaccessible due to a high population of wild animals such as leopards and sloth bears. Nevertheless, in this study we sought to highlight the site's further research potential.

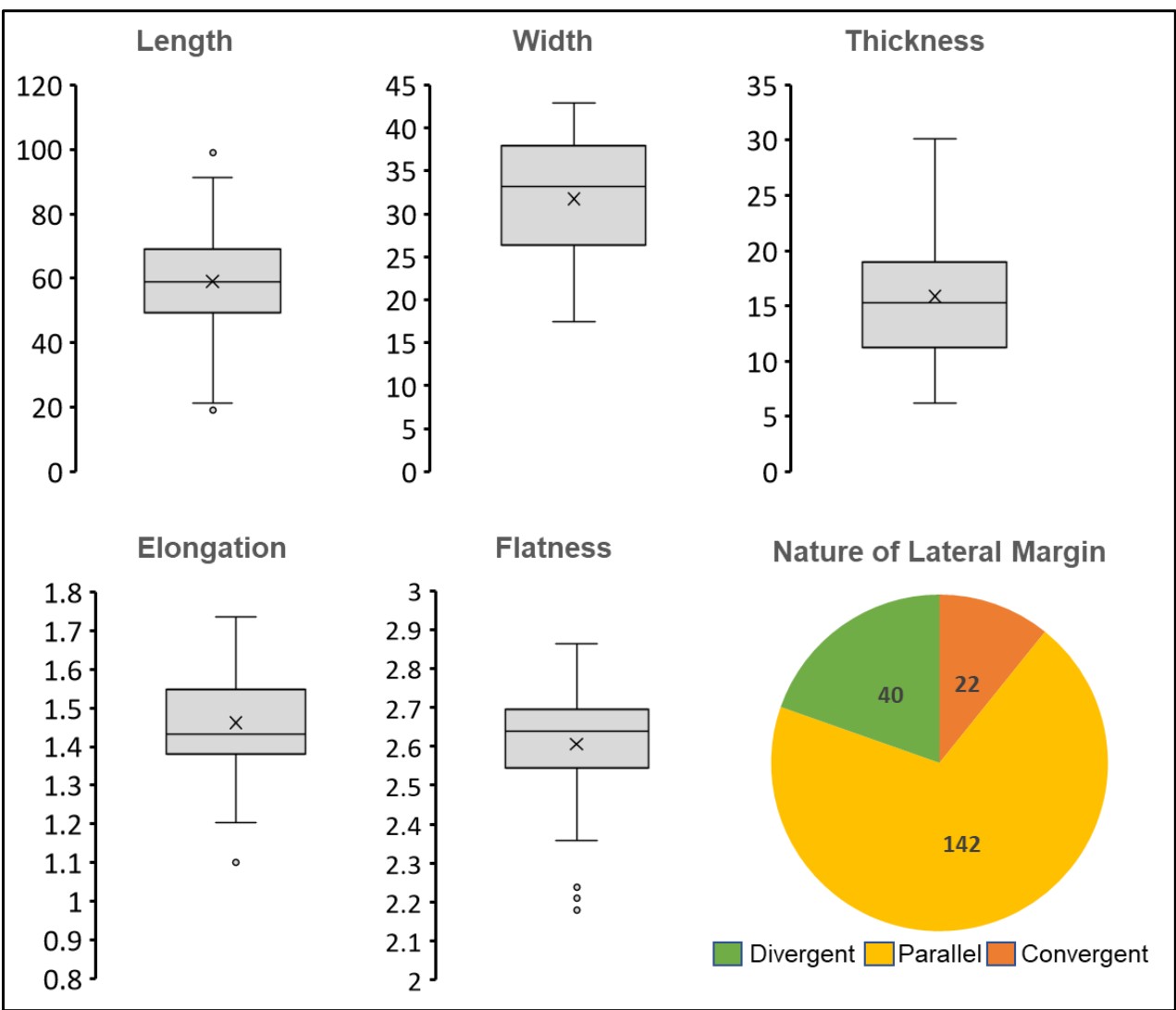

**Figure 11.** Boxplot showing different morphological attributes of BHV3 blades; pie chart explaining the nature of the lateral margin of BHV3 blades.

### 4.2.2. Typo-Technological Features

The whole assemblage is based on fine-grain quartzite. The region also yields a rich source of quartz, but it is not exploited by hominins due to its inferior conchoidal fracture. The size of the blanks (particularly blades) in BHV3 is much larger than of JGP, mostly due to differences in raw material selection. It is also reflected by the flake production efficiency (FPE). Efficiency reflects the intensity of the cutting edge per unit of raw material [23]. BHV3 blades have larger FPE than BHV3 flakes and JGP blades (Figure 12).

Constraints of raw material certainly affect the nature of blank production, which is reflected in the elongation of blades. The length of 89% of blades in BHV3 is larger than 40 mm (most of them fall in the category of 50 mm to 70 mm). The calculated mean length of the BHV3 blades is 50.69 mm with the standard deviation of 15.8 mm, which is much higher than the JGP blades. Most of the blade blanks at BHV3 are discarded primary removals. This is also visible in the length–frequency data (some of the blanks are larger than 7.5 cms). An average of 30% of the surface area of most of the blades is covered by a cortical area, which shows that these blades were part of the primary reduction sequence. The assemblage is dominated by a higher number of cores (typical cores as well as flaked clasts) and other debitage blanks. There were very few finished artefacts in the BHV3 assemblage.

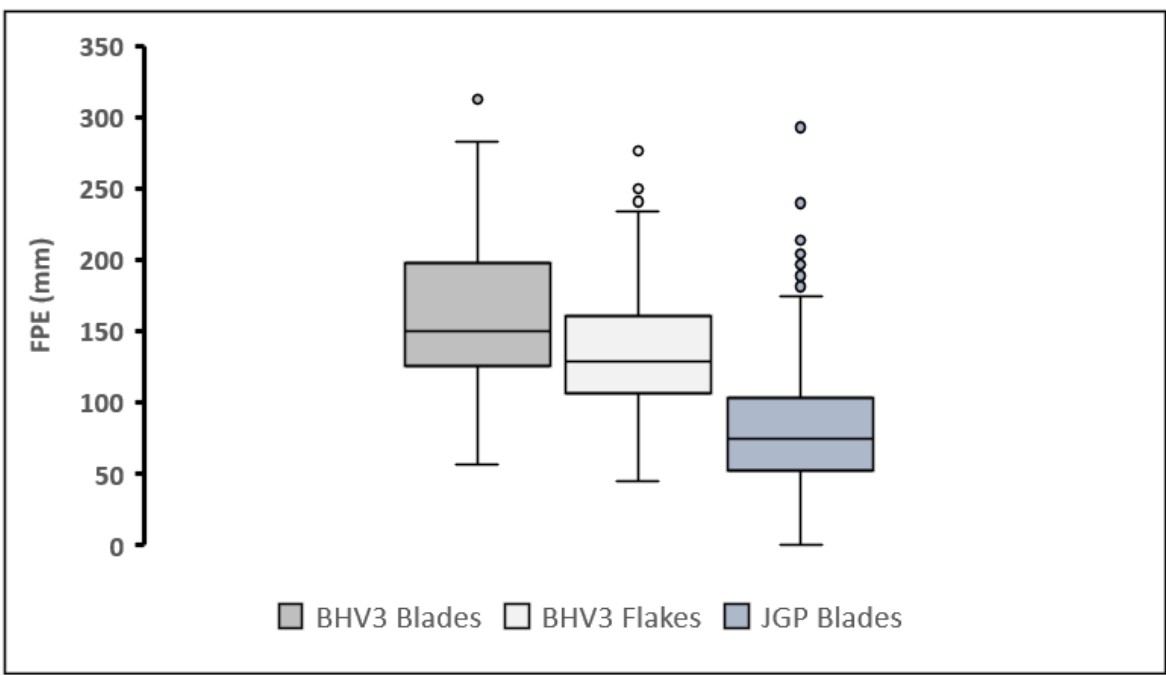

**Figure 12.** Boxplot of flake production efficiency supports the size variability that occurred due to a difference in raw material selection.

The assemblage (Table 2) reflects a contradiction in the artefact distribution, which shows a higher presence of finished artefacts. The contradiction is probably due to biases (diagnostic and finished artefacts were preferred during collection compared to other residual components) in the artefact collection. However, the predominance of parallel elongated blanks at the BHV3 assemblage possesses multiple aspects of systematic blade production at the site. A high degree of standardization in elongation and uniform thickness shows the extent of morphological standardization and knapping control.

**Table 2.** Summary of artefacts collected from BHV3.

| Type | | Quantity |
|---|---|---|
| Core | Radial/discoidal core | 8 |
| | Levallois core | 2 |
| | Blade core | 2 |
| | Amorphous core | 1 |
| Splits | Used flakes | 99 |
| | Scrapers | 46 |
| | Notch | 33 |
| | Denticulate | 3 |
| | Point | 4 |
| Total no. of artefacts | | 218 |
| Details about blade | | |
| Total no. of blade | | 106 |
| Preservations (fresh vs. abraded) | | 97–09 |
| Cortical blade | | 95 |
| Retouched blade | | 57 |
| Nature of lateral margin (divergent–parallel–convergent) | | 19-152-78-2 |

Like JGP, most of the blades at BHV3 have parallel and convergent lateral margins (Figure 11). The recurrence of parallel and sub-parallel arises on the dorsal faces of these blades shows the earlier removal of the blades. There is consistency in core preparation: the frontal face (which is also a reduction face) of most of the cores has multiple straight, parallel, or converging arises, which show the recurrent removal of blades. Along with these single platforms' unidirectional blade core, there are a high number of cores showing diverse manifestations of prepared core technologies. Like JGP, BHV3 has several PCT components (i.e., radial/discoidal, recurrent centripetal, preferential Levallois, and convergent cores). The site shows immense potential to understand technological diversification, regionality, and most importantly, transition. Future research will shed significant light on the spatio-temporal status of the assemblage at Bhuvera.

## 5. Discussion

The narrative of an Upper Paleolithic "revolution" had a strong impact on Indian paleolithic research. From Foote's Magdalenian-like assemblage of Kurnool to the famous UP site of Baghor, research into the UP was long directly linked to studying the appearance of *Homo sapiens* on the Indian subcontinent. Based on current estimates, there are 530 UP sites in the Indian subcontinent [37]. However, it is important to understand that most of the UP assemblages in the Indian subcontinent show high compositional variability [6] and do not provide a clear "revolution" or rupture in terms of technological advancement. The majority of studies consider the presence of blade technology as a major criterion to assign UP affiliation. Complete reliance on this typological approach, however, can undermine appreciation of the technological diversity of assemblages. As previously mentioned, there are several MP assemblages, such as Bhimbetka, Jwalapuram, Dhaba, etc., that yield a sizeable number of blade elements. Thus, it is crucial to reassess the technological affiliation of such assemblages.

The assemblages of JGP and BHV3 present similar examples of hybrid and diverse technological features which do not fit any straight-jacketed technology category. Assemblages at JGP and BHV3 show early signs of systematic blade production, suggesting a shift in the preference of blank size and shape. However, both assemblages also have a significant percentage of traditional Middle Paleolithic lithic components, suggesting a co-occurrence of different technological trajectories at the same point in time. Both sites present a diverse manifestation of prepared core technology. One can also see blade production as a variant within the PCT techno-complex. Associating every blade-based assemblage with laminar technology is therefore highly inappropriate. Various sites across the Old World have shown the use of PCT for systematic blade production [25,38]. It is important to understand the debitage production and overall composition of the assemblage in order to define its technological affinity, however. Current investigation suggests that blade industries should be seen as a PCT variant rather than a laminar product.

There are two probable hypotheses that can currently be formulated to understand the emergence and intensification of blade technology in South Asia.

### 5.1. Scenario 1: In Situ Emergence of Blade Technology in South Asia

As previously mentioned, there are several MP assemblages in the Indian subcontinent, such as Bhimbetka, Jwalapuram, and Dhaba, that yield an appreciable number of blade elements. Petraglia et. al. [21] and Clarkson et. al. [18,20] argue that blade technology emerged within the MP techno-complex as an adaptive response to the harsh environments of MIS 4 and MIS 3. Interestingly, blades do exist in pre-MIS 4 MP assemblages [22,37,39,40]. However, the post-MIS 4 MP assemblages show an intensification of blade production and indicate shifts in blank preference, from sub-rounded to convergent and elongated blanks. These shifts in blank preference are also observed in the MP assemblages of coastal Andhra Pradesh [40]. Additionally, the retention of bifacial components in advanced MP assemblages such as JGP and BHV shows continuation of the mode II technological practice, suggesting a deep history of South Asian MP techno-complexes.

*5.2. Scenario 2: Advent of Blade Technology and Its Diffusion with Regional MP Techno-Complexes*

The earliest occurrence of exclusive blade (including microblade) assemblages in South Asia date back to 45 ka (i.e., Site-55, Riwat) [41] and 40–48 ka (i.e., Fa-Hien lena [42], Dhaba [37] and Mehtakheri [19]). Current genetic data suggest that Anatomical Modern Humans (AMH) dispersed to South Asia by 60 ka [43,44]. Based on genetic data, Mellars [14,15] argued that AMH reached South Asia via the coastal route during MIS 4, resulting in the rapid expansion of microlithic/UP technology in the Indian subcontinent. It is possible that the newly entered population of AMH may have encountered resident populations who were practicing MP technology. There are many younger MP assemblages such as Kalpi (45 ka) [45], Dhaba (55–47 ka) [39], Jwalapuram (~34 ka) [18,20,21], and Sanghao cave (42 ka) [46], which show great temporal overlap between MP and LP assemblages. This population intermixing may have resulted in cultural diffusion and led to the formation of heterogeneous assemblage such as JGP and BHV.

## 6. Conclusions

Defining 'blade' technologies holds an important place in studies of the later episode of human origins, particularly during the late Pleistocene. Especially in South Asian context, this research area is surrounded by a series of enigma, starting with its emergence, to the rapid microlithization and intensification, and its probable relation with the expansion of *Homo sapiens*. The assemblages studied here (JGP and BHV) have all the expected components of systematic blade production, along with the presence of diminutive biface and Levallois flakes/points, showing higher variability within a single assemblage that makes it impossible to identify the populations producing these assemblages with confidence. It would be better to place JGP and BHV within the technological umbrella of the MP techno-complex (Figure 13). They show signs of technological diversification and flexible cultural preferences which are also observed in many advanced MP assemblages around the Indian sub-continent. Whether these signatures of technological heterogeneity are a result of population diffusion, local environmental response, or a combination of both are questions that future research in the region should address.

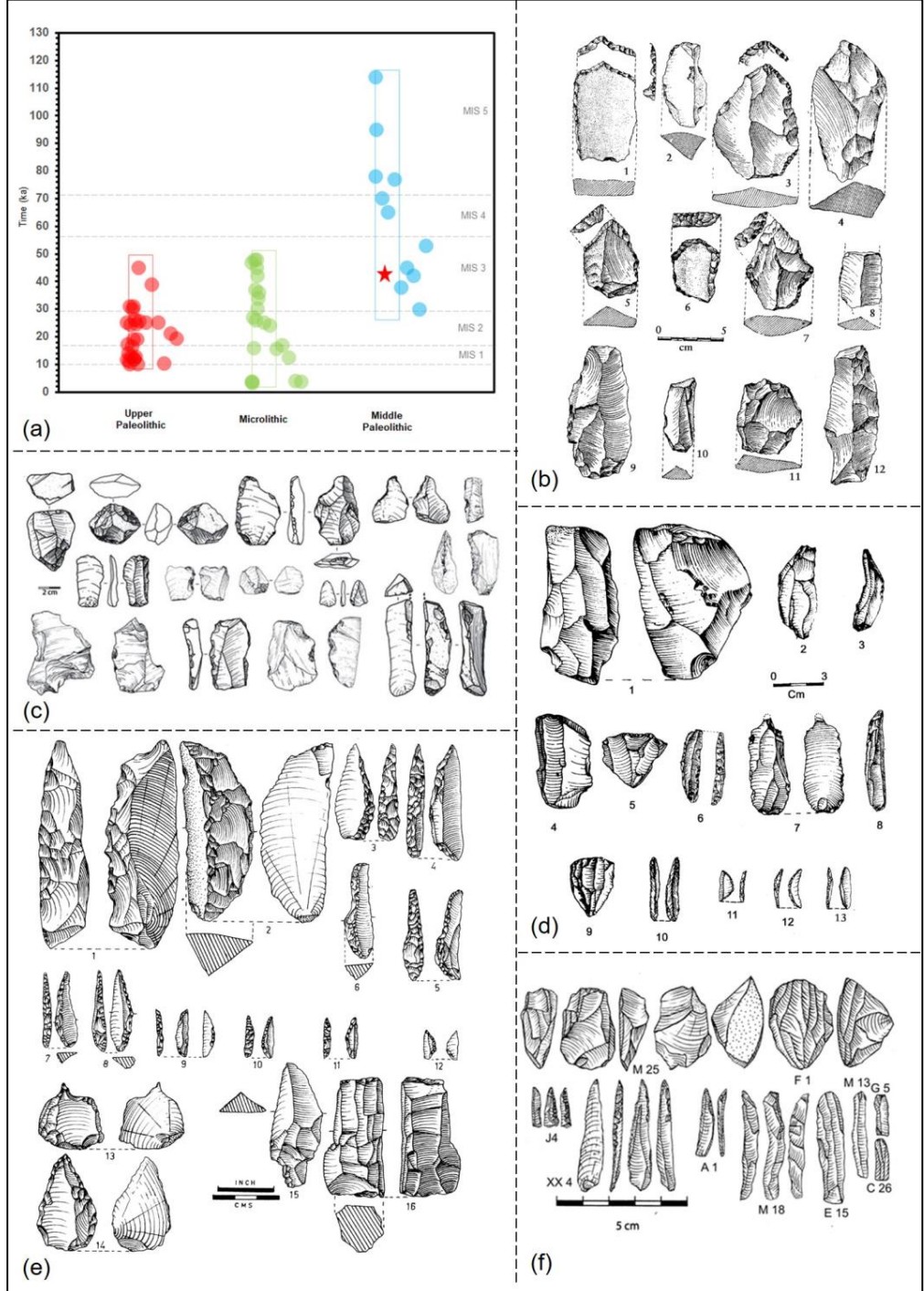

**Figure 13.** (**a**) Scatter plot showing existing absolute dates of South Asian MP, UP, and microlithic assemblages. Note the complete temporal overlap between UP and Microlithic sites. (**b**) MP assemblage of Bhimbetaka III F-23; (**c**) MP assemblage of Jwalapuram [20]); (**d**) MP (1–3) and UP (4–13) assemblages of Patne [16]; (**e**) UP assemblage of Renigunta [24]; (**f**) microblade assemblage of Mehtakheri [19]. Red star in scatter plot (**a**) represents probable temporal context of JGP and BHV assemblages. Plotted dates are collected from Murty, 1979 [16]; Mishra, 2013 [23]; and Chauhan, 2020 [37]. These illustrations are adapted from above-mentioned research papers.

**Supplementary Materials:** The following supporting information can be downloaded at: https://www.mdpi.com/article/10.3390/quat6020025/s1.

**Author Contributions:** Conceptualization—G.J. and P.A.; methodology—G.J. and P.A.; validation—G.J., P.A., V.V. and V.K.; formal analysis—G.J. and V.K.; investigation—G.J., P.A., V.V. and V.K.; data curation, G.J.; writing—G.J.; P.A., V.K. and V.V.; visualization—G.J.; supervision, P.A.; project administration, P.A. All authors have read and agreed to the published version of the manuscript.

**Funding:** This research received no external. And The APC was funded by the Max Planck Institute for School of Human History, Jena, Germany.

**Data Availability Statement:** As we mentioned, this is a preliminary study based on smaller dataset. Most of the discussed data is provided in main manuscript and Supplementary Information. Future work will provide more extensive archaeological data about JGP and BHV.

**Acknowledgments:** This work formed a part of the postgraduate project (2017–2019) submitted to the Department of Archaeology and Ancient History at the M.S. University of Baroda, Vadodara. We would like to thank multiple individuals who were involved and helped this research at different stages: Urmi Ghosh Biswas, Jesha Soni, Neha Ati, Rasagnya Mallimadugula, Navjot Kour, Nisha Das, Sourav Dasgupta, Anupama Ghosh, Disha Seth, and Katyayani Bopardikar. Without their passionate participation and input, the initial field survey could not have been successfully conducted. We would like to express our deep gratitude to the villagers of Bhuvera who assisted us during the field survey of Ratanmahal. Finally, we would like to thank Patrick Roberts for reviewing this manuscript and providing crucial inputs.

**Conflicts of Interest:** No potential or major conflict of interest is reported by the authors.

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
