# Peer review of "Assessing Systematic Blade Production in the Indian Subcontinent with Special Reference to Eastern Gujarat"

_quaternary, doi:10.3390/quat6020025_

Round 1
Reviewer 1 Report
Technically, this is a good article with very detailed technological description of the blade component. However, there is no clear presentation of geological age of the contexts. I would suggest adding the chronological information which could be presented as a Table presenting the summary on this (based on published data of the blade contexts referred in the article); this also can be done in graphic form (plotted chrono data). I believe this is extremely needed as JGP and BHV assemblages are presumed to be one of the earliest in the record potentially related to the emergence of the blade technology in the region.
Also, it is important to present stratigraphic and chronological summary for JGP locality (I assume it is available from the publication record cited by the authors). Contrary to JGP, BHV does not have it at all as I understand from the text. In that case, the authors should explain the age estimate for BHV.
I think this can improve the article which can be accepted on this basis, that is can be reconsidered after major revision.
Also, the article needs some editing.
Author Response
Authors thank the reviewer for such a comprehensive review.
Reviewer 1 expressed concern about temporal context and stratigraphic presentation of discussed sites (i.e. BHV and JGP).
Regarding chronological context:
It is to be noted, both of these assemblages are undated. The sites are situated on a hilltop that has a closed depositional context. Thus, their sedimentary matrix cannot be compared with any of the local fluvial sequence. However, the assemblage composition can be used as a relative criterion to understand probable temporal context. As suggested by reviewer 1, we added a new illustration (fig. 13) explaining probable chronological context of BHV and JGP assemblage. The illustration exhibits existing absolute ages of Upper Paleolithic (N=28), Microlithic (N=22) and Middle Paleolithic (N=11) from South Asian context. The figure of lithics from major MP, UP and microlithic assemblages from South Asia are added. As mentioned in the discussion, we believe that BHV and JGP assemblage can be defined as terminal or advanced MP assemblage like Jwalapuram 20 that are dated back to MIS 3.
Regarding stratigraphic presentation:
We thank reviewer 1 for comment on stratigraphic presentation. We have added a stratigraphic log of JGP excavation which explains the sedimentary context of JGP paleolithic assemblage. Unfortunately, we cannot provide stratigraphic log for BHV assemblage as most of the assemblages are collected from surface context. This study only reports the discovery of BHV assemblage. Future work will investigate lithostratigraphy and spatial expansion of paleolithic assemblage.

Reviewer 2 Report
The review paper is a presentation of the lithic material recovered from an open-air site. The stratigraphy presentation is quite poor, the authors can give us more information (type of sediment, relative or absolut chronology). The quality of the photographs in some figures is of low quality. I am aware that it is not easy to photograph quartz tools but there are methods to remedy these deficiencies (light control, mark the extractions as you have already done in other figures). Regarding the discussion and the conclusions, these are very summarized. The authors could place more emphasis with comparisons with deposits of the same coronalogy in the area. The conclusions should collect the main ideas and results of the paper. Check line spacing, there are paragraphs with double space and others with single space.
Author Response
Authors thank the reviewer for such a comprehensive review.
Reviewer 2 expressed concern about temporal context, stratigraphic presentation of discussed sites (i.e. BHV and JGP), and manuscript formatting-editing.
Regarding stratigraphic presentation:
We thank reviewer 2 for comment on stratigraphic presentation. We have added a stratigraphic log of JGP excavation which explains the sedimentary context of JGP paleolithic assemblage. Unfortunately, we cannot provide stratigraphic log for BHV assemblage as most of the assemblages are collected from surface context. This study only reports the discovery of BHV assemblage. Future work will investigate lithostratigraphy and spatial expansion of paleolithic assemblage.
Regarding chronological context:
It is to be noted, both of these assemblages are undated. The sites are situated on a hill-top that has a closed depositional context. Thus, their sedimentary matrix cannot be compared with any of the local fluvial sequence. However, the assemblage composition can be used as a relative criterion to understand probable temporal context. As suggested by reviewer 1, we added a new illustration (fig. 13) explaining probable chronological context of BHV and JGP assemblage. The illustration exhibits existing absolute ages of Upper Paleolithic (N=28), Microlithic (N=22) and Middle Paleolithic (N=11) from South Asian context. The figure of lithics from major MP, UP and microlithic assemblages from South Asia are added. As mentioned in the discussion, we believe that BHV and JGP assemblage can be defined as terminal or advanced MP assemblage like Jwalapuram 20 that are dated back to MIS 3.
We apologize for not being thorough in our proof-reading process. All the comments that are made here are appropriate and have been taken into account during revision. The relevant changes can be seen in the revised manuscript.

Reviewer 3 Report
The problems you address are real and interesting. However, the fact that in both cases the undatable assemblages come from surface collections and are deeply disturbed by taphonomic processes limits the value of what you presented. I feel that you should make more clear this kind of handicap, particularly because you are working with white quartz, notoriously a hard bone for lithic studies. The english is generally poor and I found difficult to understand not a few sentences. I enclose your file with comments and critical points in yellow - I hope this will help.

Author Response
Authors thank the reviewer for such a comprehensive review.
We apologize for not being thorough in our proof-reading process. All the comments that are made here are appropriate and have been taken into account during revision. The relevant changes can be seen in the revised manuscript.

Round 2
Reviewer 1 Report
I appreciate your effort to clarify my questions on the stratigraphy and age estimate. In present form, your description gives much better idea on the material used in the study.
Author Response
Authors thank the reviewer for such a comprehensive review.
Reviewer 3 Report
Dear authors,
I have seen the changes you made to the first version and they work, the article is much more clear and useful. My personal dissatisfaction remains with the substandard drawings of Fig. 5 (drawing not made by an experienced professional specialist, no lateral view so that there is no way of assessing the thickness of the tools). The other figures of the lithics are not homogeneous (photographs with black lines for marking the detachment surfaces, and other without black lines, so that there are three different forms of illustration of lithics in the same paper and this creates confusion. At least, you should explain that the tools of Fig. 5 are made with a different base material, and insist a little on the constraints imposed on the reproduction of the tools by macro-crystalline quartz.
Author Response
Authors thank the reviewer for such a comprehensive review.
Regarding lithic illustrations:
The authors accept reviewer’s recommendation and have added the lateral views of the stone tools in figure. 5. The authors would like to mention that it is immensely difficult to do scar patterning of vein-quartz artefacts. However, we have replaced figure 7 with the line drawing of JGP blades (showing dorsal, ventral, and lateral faces of blades).
Regarding description of PCT products in figure 5:
It is to be noted that the current work solely focuses on the blade implements of JGP assemblage that are completely made of vein quartz. The main intent of providing figure 5 is to showcase the assemblage’s heterogeneity (we’ve discussed the same in paragraph 3.1.3). We completely agree with the reviewer that practicing Levallois technology on raw material like vein-quartz may have presented several challenges to knappers. We have provided a brief explanation about this in the paragraph 3.1.4.
However, we believe elaborate geometric morphometric analysis of MP assemblage (based on vein-quartz as well as quartzite) of JGP will help us to explore raw material induced technological variability at the site of Jogpura. Authors intend to conduct such studies in future.
